

# Multi-objective calibration of the Community Land Model Version 5.0 using in-situ observations of water and energy fluxes and variables

Tanja Denager[1], Torben O. Sonnenborg[2], Majken C. Looms[1], Heye Bogena[3] and Karsten H. Jensen[1]

[1] Department of Geosciences and Natural Resource Management, University of Copenhagen, Øster Voldgade 10, 1350 Copenhagen, Denmark

[2] Geological Survey of Denmark and Greenland, Øster Voldgade 10, 1350 Copenhagen, Denmark

[3] Agrosphere Institute, IBG-3, Forschungszentrum Jülich GmbH, 52428 Jülich, Germany,

*Correspondence to*: Tanja Denager (tad@ign.ku.dk)

**Abstract:** This study evaluate water and energy fluxes and variables in combination with parameter optimization of the state-of-the-art land surface model Community Land Model version 5 (CLM5), using six years of hourly observations of latent heat flux, sensible heat flux, groundwater recharge and soil moisture.

The results show that multi-objective calibration in combination with truncated singular value
decomposition and Tikhonov regularization is a powerful method to improve the current practice of using look-up tables to define parameter values in land surface models. Furthermore, reliability of the optimized model parameters can be estimated by statistical measures such as identifiability and relative error variance reduction. As in most other eddy covariance studies, closure of the land surface energy balance is not achieved on observation data. However, using direct measurement of
turbulent fluxes as target variable, the parameter optimization is capable of matching simulations and observations of latent heat, especially during the summer period, while simulated sensible heat is clearly biased. The fact that CLM5 is not capable of matching sensible heat, not even with advanced parameter optimization of model parameter values, suggests that the lack of energy closure is due to biases in the sensible heat flux. The results from this study contribute to
improvements in model characterization of water and energy fluxes. It is underlined that parameter calibration using available observations of hydrologic and energy fluxes and variables is necessary to obtain the optimal parameter set of a land surface model.



# 1 Introduction

Hydrological processes play a fundamental role in land surface water and energy cycles. A land surface model (LSM) is a tool for linking energy and water processes at the land surface and is used to study and understand the processes controlling the transport of energy and water. There is a need for evaluating the hydrologic performance of LSMs based on comprehensive in-situ data on water and energy fluxes and variables. Climate change and changes in land-use/land cover further increase the demand for quantification of the water and energy fluxes and for investigating the predictive capability of LSMs (Clark et al., 2015; Dai et al., 2003; Oleson et al., 2008; Overgaard et al., 2006).

LSMs simulate the vertical water and energy fluxes from the top of the canopy, through the canopy and stem, through the root zone and down to the groundwater table. The vertical fluxes and states are simulated based on coupled flow and energy equations subject to various boundary conditions and described by a large number of parameters. It is common practice is to use lookup tables to define a-priori parameter values (Hou et al., 2012; Rosero et al., 2010). However, many LSM components are based on relatively few observations and idealized laboratory experiments (Stöckli et al., 2008), and existing LSMs are generally not tested on in-situ hydrological observational data (Clark et al., 2015). Thus, LSMs are typically under-constrained (De Lannoy et al., 2011; Stöckli et al., 2008), and their capability for hydrological simulations at watershed scales has not been adequately studied (Li et al., 2011). It is standard practice in LSMs that a-priori assignment of parameter values is based solely on vegetation type or soil texture. However, several authors suggest that the parameterization in LSMs should also consider the climatic conditions (Rosero et al., 2010), as local climate has an important impact on the parameter values, especially when realistic hydrological responses should be captured (Huang et al., 2013).

Many LSM studies focus on continental to global effects, while hydrological model studies often have a catchment-based focus. With the development of hydrological observatories (Bogena et al., 2018), critical zone observatories (Guo and Lin, 2016), FLUXNET (Chen et al., 2018; Wilson et al., 2002) and similar observational programs, more and more attention is paid to hydrological performance of LSMs at local and regional scales (Carrillo-Rojas et al., 2020; Stöckli et al., 2008). It is important to test and evaluate LSMs at point scale to assess their predictability and their usefulness in global simulations (Dai et al., 2003). However, the smaller scale models are also highly relevant as they represent the scales at which societies make decisions. LSMs are used to inform and support natural resource management, for example, by estimating the evapotranspiration components of various land-covers and hereby provide a platform for water and land use management under current and future climate conditions.



LSMs are simplified representations of the landscape and many of the parameters of the process relations cannot be directly measured (Gupta et al., 1999). Additionally, there are extensive structural differences among LSMs (Clark et al., 2015). Therefore, the majority of parameters in LSMs are often model dependent and hence difficult to transfer and compare between different

LSM schemes (Rosero et al., 2010).

Over time LSMs have been further developed to address a broad range of terrestrial ecosystems related scientific questions (Lawrence et al., 2019a), e.g. cycling of energy, water, carbon and nitrogen. The "bewilderingly large set of processes" (Clark et al., 2015) incorporated into LSMs heavily increase model complexity and the associated number of parameters that governs the model

equations, which emphasizes the need for parameter estimation and performance evaluation (Mendoza et al., 2014). Some of the commonly used LSMs are ORCHIDEE (Krinner et al., 2005), CLM (Dai et al., 2003) and NOAH-MP (Niu et al., 2011). Advanced calibration techniques are widely used in hydrology for parameter estimation including techniques to quantify uncertainties. Contrary to hydrological modeling, calibration of LSMs is relatively uncommon (Davison et al.,

2016; De Lannoy et al., 2011), and only a limited number of studies have dealt with calibration and sensitivity analysis of the energy and hydrology parameters in LSMs (Gupta et al., 1999; Pauwels and De Lannoy, 2011). Examples are: i) Rosero et al. (2010), who quantified the parameter sensitivity of both soil and vegetation parameters using the Sobol's method in the Noah LSM by minimization of a RMSE multi-objective criteria of sensible heat flux (H), latent heat flux (LE),

ground heat flux (G), soil temperature ($T_s$) and soil water content (SWC); ii) (Pauwels and De Lannoy, 2011) also combined energy fluxes, as well as SWC, when calibrating a simple water and energy balance model using the spectral domain method; and iii) Davison et al. (2016) performed a single-objective calibration on streamflow and concluded that the simulation of streamflow clearly has an influence on the simulated LE.

Several studies have carried out sensitivity analyses on former versions of CLM. (Göhler et al., 2013) used eigendecomposition in a sensitivity study of 66 parameters in CLM3.5 using measurements of energy fluxes and photosynthesis, while both Huang et al. (2013) and Sun et al. (2013) performed sensitivity analyses using satellite-based LE estimates and daily streamflow measurements, respectively, for evaluating the sensitivity of hydrologic parameters in CLM4.0.

Hou et al. (2012) made an uncertainty quantification using the quasi-Monte Carlo approach to evaluate the sensitivity of LE and H to the hydrological input variables in CLM4.0 and Jefferson et al. (2016) used energy fluxes in the active subspaces method to evaluate parameter sensitivity in the ParFlow-Community Land Model. Zhang et al. (2017) calibrated soil texture parameters using data assimilation methods and observed SWC. Hence, previous studies have shown that both





energy and hydrological fluxes and variables are sensitive to the parameterization of a CLM, emphasizing the need for parameter optimization.

In this study, we evaluate in-situ water and energy fluxes and variables at an agricultural field site in Denmark. We apply the state-of-the-art LSM Community Land Model version 5 (CLM5). This
recent version of CLM includes a wide range of modifications in its structure and parameterization over previous CLM versions (Lawrence et al., 2019a). Only few calibration studies for CLM5 have been reported (Dombrowski et al., 2022), however, through their validation of CLM5, Cheng et al. (2021) state that calibration of hydrologic parameters are needed to improve simulations of subsurface runoff. Recently, Dombrowski et al., (2022) performed a sensitivity analysis using the
prognostic crop module in CLM5.  We calibrate a point-scale CLM5 against observations of LE, H, recharge (q) and SWC from the Danish hydrological observatory HOBE (Jensen and Refsgaard, 2018) using well-established calibration methods in hydrology. Our observation dataset is exclusive in the way that we have all observations are available for closing the long-term water and energy balance at point-scale including groundwater recharge measurements, which have not
previously been used for evaluating and calibrating a LSM. We apply a methodological approach that combine (1) multi-objective calibration, (2) truncated singular value decomposition and (3) Tikhonov regularization, by using the PEST program suite (Doherty, 2015). After the auto-calibration, we evaluate the model parameter uncertainty by means of identifiability and relative error variance reduction (Doherty and Hunt, 2009).



## 2   Methods

### 2.1   Study site

The Voulund site is an agricultural field observatory (Jensen and Refsgaard, 2018) located in a temperate climate in the western part of Denmark on flat terrain. During the study period, the field

was cultured with rotations of spring and winter barley, with grass-species as cover crop during the autumn and winter season. The ploughed root zone of 30 cm contains approximately 4.5% organic matter (Andreasen et al., 2020), while there is little organic matter content below 30 cm. The soil is sandy with only very little clay content (Vasquez, 2013). The field site is a part of the Danish Hydrological Observatory (HOBE).

Hourly forcing data from the period 2010-2015 were used for the analysis. From a flux tower, measurements of energy fluxes were obtained (Ringgaard et al., 2011) and the tower was also equipped with sensors of temperature, relative humidity and radiation components. Wind speed and atmospheric pressure were obtained from a meteorological station. The precipitation dataset is constructed based on observations from six undercatch-corrected precipitation gauges (Denager et

al., 2020). Recorded irrigation amounts are included as additional precipitation in the precipitation dataset. Soil temperature ($T_s$) were obtained from two capacitance sensors located right below soil surface.

To evaluate the performance of the CLM5 model we used measurements of LE, H, q and SWC in the top soil layer (0-20 cm). Four percolation lysimeters measured recharge q (Schelde et al., 2011)

and measurements of SWC in the top soil was obtained from a cosmic ray neutron sensor (CRNS) (Andreasen et al., 2020; Bogena et al., 2022). Two heat flux plates measured ground heat flux (G) at 0.05 meter below ground level (mbgl). Net radiation (Rn) was calculated as the difference of incident and reflected shortwave (Sin - Sout) and longwave radiation (Lin - Lout) summed. In Denager et al. (2020) further details on site characterisations and data collection are provided.

### 2.2   Model description

The open-source LSM Community Land Model version 5 (CLM5) (Lawrence et al., 2019a, 2019b) is the land component of the Community Earth System Model (CESM), and it simulates the soil-plant-atmosphere exchange processes. We applied this process-based model in single-point mode, uncoupled from the climate model and driven by hourly in situ site-specific climate forcing data.

We used the original and publicly available release code of CLM5 with the modifications mentioned below.

CLM5 includes biophysical, biochemical, ecological and hydrological processes that are described by equations with a large number of parameters. Thermal and soil hydraulic parameters are



estimated with built-in pedo-transfer functions from simple soil properties such as soil texture (fractions of sand and clay) (Nachtergaele et al., 2009) and soil organic carbon (Lawrence and Slater, 2008). CLM5 simulates unsaturated flow by the one-dimensional Richards' equation for vertical flow and surface runoff based on a TOPMODEL-based parameterization (SIMTOP) (Niu

et al., 2007, 2005). Surface water storage is simulated as a function of microtopography (Lawrence et al., 2019a). The soil column is divided into 20 hydrological active soil layers (0-8.6 mbgl) (Lawrence et al., 2019a), and with the thickness of each layer increasing from top to bottom. While CLM5 calculates water flux and SWC for all 20 hydrological active layers, it is assumed that the soil texture is homogeneous within each of two horizons; the root zone (0-0.32 mbgl) and below

root zone (0.32 – 8.6 mbgl). In the present application of CLM5, the simulated groundwater recharge $q_{sim}$ is found as the water reach the bottom of the eleventh soil layer, corresponding to the depth of the bottom of the lysimeters. In this study, we compare the average SWC of CLM5 layers 1 to 4 (0-20 cm) with the SWC measured by the CRNS, which corresponds to the average CRNS measurement depth at the site. All simulations were carried out with hourly time steps covering the

period 2010-2015. Simulated recharge and soil water content are compared to the outflow from lysimeters and CRNS estimated SWC, respectively.

The lower boundary condition of the model was a water table head-based boundary (https://www.cesm.ucar.edu/models/cesm2/settings/current/clm5_0_nml.html). This modification was needed as default CLM5 settings of the lower boundary condition raised the groundwater table

above the level of the bottom of the lysimeters.

CLM5 was applied in satellite phenology mode (CLM5-SP), where the carbon and nitrogen biogeochemistry cycles were deactivated and plant phenology was represented by leaf area index (LAI), stem area index (SAI) and canopy height (height_top). LAI is the green area index, while SAI includes dead leaf and litter.

The energy fluxes considered in CLM5 include direct and diffuse short-wave radiation as well as absorbed, transmitted, and reflected longwave radiation by soil and vegetation. CLM5 simulates the turbulent fluxes of H and LE numerically through the Monin-Obukhov similarity theory (Lawrence et al., 2019a), which relates the turbulent fluxes to the differences of mean temperature and humidity (Wang and Dickinson, 2012). CLM5 calculates many individual processes. For

example, soil evaporation, canopy evaporation and transpiration are parameterized individually, and the sum of these individual component terms makes up total $E_{sim}$. A detailed description of the CLM5 framework is available in Lawrence et al. (2019a).

Energy is conserved at every time step (Lawrence et al., 2019a):





$$Rn_{CLM} = H_{CLM} + LE_{CLM} + G_{CLM} \quad (1)$$

where $Rn_{CLM}$ is the net radiative flux, H is the sensible heat flux, LE is the latent heat flux and G is the ground heat flux. CLM5 simulates LE and H explicitly based on physical laws, while G is considered a residual term for closing the energy balance (Lawrence et al., 2019a). This approach for closing the land surface energy balance is used in the majority of the available LSMs (Kracher et al., 2009).

The uniform vegetation and topography around the study site justifies the assumption of the land surface domain as a single grid cell implying that only vertical variations in energy and water fluxes are considered. The assumption of homogeneity simplifies the model considerably.

A spin-up configuration enables CLM5 to reach a quasi-equilibrium state prior to simulation period of interest. 1000 years of spin-up were used from cold start with the described modifications of the model setup, and four years (2012-2015) of forcing data were recycled to achieve proper initial conditions. Additionally, since the calibration process changes the model behavior through parameter adjustments, we included four years spin-up preceding each of the final simulations for
the complete period of six years.

CLM5 differentiate between "*surface runoff*" from the SIMTOP runoff model (Niu et al., 2005) and "*surface water runoff/surface water storage*" based on microtopography (Lawrence et al., 2019a). In SIMTOP precipitation that falls over the saturated fraction of a grid cell is immediately converted to surface runoff. Surface runoff at the study site is almost absent. Therefore, maximum
possible saturated area fraction ($F_{max}$) was set to zero resulting in nonexistent surface runoff.

Meteorological forcing data include precipitation, air temperature, wind speed, surface air pressure and relative humidity, while radiation forcing data includes incident solar (Sin) and incident long-wave radiation (Lin).

As the intension was to calibrate CLM5 outputs against observed flux data, it is of critical
importance that the specified $Rn_{obs}$ is in agreement with $Rn_{sim}$. We identified systematic errors in the measurements of absolute longwave radiation components. However, although the values of absolute longwave radiation were inaccurate, we assume that the difference between $Lin_{obs}$ and $Lout_{obs}$ was reliable, and thus assuming that $Rn_{obs}$ calculated as $Rn_{obs} = Sin_{obs} - Sout_{obs} + Lin_{obs} - Lout_{obs}$ will represent the net radiation at the field site.

$L_{in}$ specified to the model was computed as a differential term because CLM5 computes $L_{out}$ from Stefan-Boltzmann's law (Stöckli et al., 2008):



$$L_{in} = R_n - S_{in} + S_{out} + \sigma \left( \frac{T_a + T_s}{2} \right)^4 \qquad (2)$$

where Rn, Sin and Sout are net radiation, incoming solar radiation and outgoing solar radiation respectively (W m$^{-2}$), σ is the Stefan-Bolzmann constant (5.67·10$^{-8}$ Wm$^{-2}$ K$^{-4}$), $T_a$ is the air temperature (K) and $T_s$ is the soil-surface temperature (K).

In the calibration we used six different observation data sets as optimization targets; Rn, Sout, LE, H, q and SWC in the top soil layer, all in hourly resolution. We considered eleven individual scenarios (A-K) where calibration was carried out against different combinations of observation data types (Table 1). The scenarios were designed to both study the worth of hydrological data in an energy based LSM, and to study the reliability of LE and SH observations, respectively. Rn and

Sout were included as optimization targets to ensure persistent match between observations and simulations of Rn and Sout.

The observed energy fluxes do not meet long-term energy balance closure (Denager et al., 2020). Many studies introduce corrections of the observed energy fluxes LE and H to meet energy balance closure (Carrillo-Rojas et al., 2020; Chen et al., 2018; Davison et al., 2016). Such a correction of

the observed turbulent fluxes was not applied here as the goal specifically was to analyze the energy balance components using CLM5.

## 2.3    Calibration approach

Calibration is a challenge when models are complex and the number of parameters is high (Doherty et al., 2010). We applied the PEST suite programs (Doherty, 2018a, 2018b) to calibrate CLM5.

PEST is an open source software and model-independent, and provides highly parameterized inversion and model parameter uncertainty analysis (Doherty et al., 2010).

In mathematical regularization using the subspace method, the parameter space is divided into a solution space and a null space. The solution space comprises combinations of parameters that can be estimated uniquely from the available observations, while the null space includes parameters

combinations that cannot be estimated on the basis of the observations. Truncation of low singular values provides a threshold between solution and null spaces (Doherty et al., 2010).

Focus was given to a set of 30 time-invariant model parameters (Table 1 and Appendix), chosen for their direct mechanistic impacts on responses of energy and water fluxes.  To keep the analysis simple, we decided to include only parameters represented in look-up tables and to disregard hard-

coded parameters, parameters determining pedo-transfer functions as well as parameters influencing e.g. snow hydrology. We kept all these parameters at the prescribed values.



Regularization converts an ill-posed problem to a well-posed problem and prevents overfitting. Truncated singular value decomposition identifies insensitive or highly correlated combinations of parameters and excludes them from the calibration (Doherty, 2015) and through Tikhonov regularization we honored the observed parameter values and a-priori information from look-up tables, as those were given as the prior-knowledge/initial values (Table 1 and Appendix).

In CLM5 the soil and hydraulic parameters including porosity, saturated hydraulic conductivity and the Clapp-Hornberger exponent B in the functional relationships for retention and unsaturated hydraulic conductivity are derived from soil texture (percentage of sand/clay and organic) in each soil layer (Lawrence et al., 2019a) using built-in pedo-transfer functions. Measured soil texture were used as prior-knowledge/initial values (Vasquez, 2013). Those were slightly different from look-up table parameter values (Table 1 and Appendix). The soil carbon density in the root zone was fixed at a value of 6 kg/m$^3$, since this entails an organic matter content corresponding to the measured value of 4.5% (Andreasen et al., 2020). Soil colour determines dry and saturated soil albedo (Fisher et al., 2019). Soil colour was not included in the calibration because the parameter estimation tool was not able to handle parameter values as integers. The look-up parameter value of soil colour for the field site is 13; we used this value in the simulations.

The a-priori satellite-derived *LAI* and *SAI* values were aggregated from high resolution input datasets (Cheng et al., 2021). According to our basic knowledge of the field site (Herbst et al., 2011) the a-priori LAI values as derived from satellite images seemed rather small. Therefore we used initial values for LAI assessed from Herbst et al. (2011). We included all 12 monthly LAI parameters in the calibration. We used the SAI values from the look-up table and did not include them in the optimization. Initial values of the eight optical properties parameters were defined according to the look-up table values.

We used the single Plant Functional Type (PFT) "C3 Unmanaged Rainfed Crop" (Lawrence et al., 2019a) as a-priori vegetation parameter values. The prescribed leaf/stem orientation index for "C3 Unmanaged Rainfed Crop" of -0.3, was changed to -0.5, as this is the prescribed value for spring wheat (Lawrence et al., 2019a).

Parameter limits were given wide intervals to give full freedom to the parameter optimization. Prior calibration parameter variability ($\sigma_{i\,pre}$) was given as a standard deviation of 0.5 in the log space of the respective parameters.

The multi-objective function ($\phi_{observation}$) that is minimized by PEST is defined as the squared sum of weighted residuals.



$$\varphi_{observation} = \sum_{i=1}^{m} \sum_{j=1}^{n} \left( \omega_{y,i,j} \left( y_{obs,i,j} - y_{sim,i,j} \right) \right)^2 \qquad (3)$$

where m is the number of observation groups in the given optimization, n is the number of respective Rn, Sout, LE, H, q and SWC observations, ω is the weight of the observations, $y_{obs}$ and $y_{sim}$ are observed and simulated values, respectively. We ensured uniform weighting between the

different observation groups to avoid single observation group to excessively dominate the parameter estimation.

Regularization was introduced in all calibrations by adding the regularization objective function ($\phi_{regularization}$) to $\phi_{observation}$. As we used preferred value regularization, $\phi_{regularization}$ consists of the weighted least squared of the difference between parameter value and preferred (a-priori)

parameter values. The total objective function ($\phi_t$) thus comprises the sum of the observation and the regularization objective functions

$$\varphi_t = \varphi_{observation} + \mu^2 \varphi_{regularization} \qquad (4)$$

were μ is the weight factor of the regularization objective function (Doherty, 2018a).

In mathematical regularization we seek an "appropriate" fit, rather than the best possible fit

between simulations and observations (Doherty et al., 2010). An acceptable fit is specified by PHIMLIM, which defines a threshold value that the observation objective function must not fall below. Hereby, a balanced optimization is obtained with respect to observations and prior parameter values. PHIMLIM was set 10% higher than the lowest achieved objective function, and PHIMACCEPT another 10% higher than PHIMLIM as recommend by Doherty (2018a).

The weights to the individual observations were assigned such that they were proportional to the standard deviation associated with the observation. The standard deviation was assumed to be 10% of the absolute observation value. To ensure that all observation time steps had a balanced impact on the objective function, we developed a simple model of the observation weights of LE, H and q. Hereby, larger observations are given a higher weight than smaller observations and time steps

where $y_{obs} \approx 0$ is prevented from having inappropriate high weight and therefore inappropriate high impact on the objective function.

$$\omega_i = \frac{1}{a - 0.1 \cdot |y_{obs}|} \qquad (5)$$

where $a_{LE} = 1000$, $a_H = 1000$, $a_{Rn} = 1000$ and $a_q = 1$. All SWC observations were given the same weight and thus not dependent on the observation value.



All calibration scenarios were assessed based on mean error (ME), mean absolute error (MAE), root mean square error (RMSE), Nash-Sutcliffe coefficient (NSE) and Pearsons correlation coefficient (r) for each of the four observation groups LE, H, Rn, q and SWC.

$$ME = \frac{1}{n}\sum_{j=1}^{N}(obs_j - sim_j) \qquad (6)$$

$$MAE = \frac{1}{N}\sum_{j=1}^{N}|obs_j - sim_j| \qquad (7)$$

$$RMSE = \sqrt{\frac{1}{N}\sum_{j=1}^{N}(sim_j - obs_j)^2} \qquad (8)$$

$\quad NSE = 1 - \dfrac{\sum_{j=1}^{N}|obs_j - sim_j|^2}{\sum_{j=1}^{N}|obs_j - \overline{obs}|^2} \qquad (9)$

$$r = \frac{\sum_{j=1}^{N}(obs_j - \overline{obs}) \cdot (sim_j - \overline{sim})}{\sqrt{\sum_{j=1}^{N}(obs_j - \overline{obs})^2 \cdot \sum_{j=1}^{N}(sim_j - \overline{sim})^2}} \qquad (10)$$

where N is the number of observations in the given observation group. All summary statistics were
calculated on hourly time basis.

A small ME suggests that the overall model fit is not biased, however, positive and negative errors may cancel out implying that ME may be a weak indicator of the goodness of model fit. Instead, MAE may be a better indicator of the model performance. RMSE is a performance criteria, which gives higher weight to large errors as opposed to MAE that weights all residuals equally. The
innate character of RMSE is very much related to the objective function. NSE and r are both unit less, should ideally be as close to 1 as possible and comparable across data types. NSE is a measure of the model's ability to match the temporal variability, while r is a measure of the strength of the linear relationship. For ME, MAE and RMSE the closer the metrics are to 0, the better the model performs.  The optimized fluxes and states of the system is evaluated through those six metrics
(including objective function). It is important to keep in mind that the optimization tool PEST uses the objective function and that this does not necessarily improve all other metrics.



Aside from parameter estimation, the PEST software package contains a collection of utility programs for calculation of the model parameter uncertainties developed under the assumption of linearity. Thus, the uncertainty estimates are approximates, but can nevertheless provide useful information even though the system may violate the assumptions (Doherty, 2015). The truncation point (or threshold) between the null and solution space is a generic mathematical concept that enable an investigation of model error (Doherty, 2015; Doherty et al., 2010).

To assess the parameter importance, we used the two statistics "identifiability" and "relative error variance reduction" (Doherty and Hunt, 2009) calculated by the PEST utility programs IDENTPAR and GENLINPRED. These statistics are based on the same concepts as those applied by mathematical regularization and rely on singular value decomposition of a weighted sensitivity matrix. Opposite to the one-at-a-time sensitivity analysis approach, the identifiability and relative error variance reduction determine the significance of the parameters while taking the interactions among them into account (Doherty and Hunt, 2009).

The identifiability expresses to which extent a parameter can be estimated uniquely based on the extent that the parameter is located in the solution space and hence how much it is informed by available observation data. When the identifiability of a parameter is 0, the dataset possesses no information with respect to that parameter, and the uncertainty is not reduced through the calibration process. When the identifiability of a parameter is 1, it does not mean that the parameter can be estimated without error, but it indicates that all of its potential for errors are dominated by and originates from the noise of the observation data (Doherty and Hunt, 2009).

The relative error variance reduction ($r_i$) describes to which extent the calibration process reduces the variance of a parameter from the pre-calibration level (Doherty and Hunt, 2009).

$$r_i = 1 - \frac{\sigma_{i\ post}^2}{\sigma_{i\ pre}^2} \qquad (11)$$

where $\sigma_{i\ post}^2$ is post-calibration error variance associated with estimation of parameter i, and $\sigma_{i\ pre}^2$ is its pre-calibration error variance assigned by expert knowledge (Table 1 and Appendix).





## 3   Results

To provide a basis for comparison we ran a control simulation using CLM5's a-priori (look-up table) parameter values (Scenario X). Additionally, a simulation was run (Scenario Z) where some look-up table parameters values were replaced by observed parameter values. Table 1 present the soil texture parameters and the plant functional type (PFT) parameters. LAI and optical parameters can be found in Appendix. Look-up table and initial parameter values are listed together with the optimized parameters for all calibrated scenarios. Scenarios A, E and I are calibrations with LE, H and q, respectively, as targets., The remaining scenarios are multi-objective calibrations using different combinations of observation data types. The summary statistics are given in Fig- 1 where the top row show the initial and control runs together with statistics on the observed data. In row no. 2 calibration results using LE as targets plus LE combined with other measurement types as targets are presented. Row no. 3 is similar to row no. 2, where LE is substituted by H. In the last row results using different combinations of targets is shown.

Table 1

Figure 1

Figure 2

Figure 3

Figure 4

As $Rn_{obs}$ was used indirectly to obtain incident longwave radiation for model forcing (Eq. 2), there is a good match between both $Rn_{obs}$ and $Rn_{sim}$, and $Sout_{obs}$ and $Sout_{sim}$, in the control run. To ensure that simulated and observed Rn and Sout agree in the optimization process, Rn and Sout were included in the objective function (Eq. 3) and given the same group weight as for the other variables in the objective function. We included Rn and Sout in the objective function to ensure accordance between observed and simulated Rn and the short wave radiation components. It is important to note that in the control run and initial model run (Scenario X and Z) an excellent match between $Rn_{obs}$ and $Rn_{sim}$ was already obtained and therefore we do not expect the metrics for Rn to improve in the calibrated scenarios (Fig 1).

### 3.1   Analysis of the control run

Simulations based on look-up parameter values for the field site (Scenario X) highly overestimate daily H all year except in July and August (Fig. 2b). On the contrary, LE is underestimated during the cold season from September to April, especially in March and April (Fig. 2a). This model conceptualization fails to reproduce the correct partitioning between LE and H during the grain



filling and harvest period in July and August, where LE is highly overestimated (Fig. 2a) and H underestimated (Fig. 2b).

Regarding the unsaturated zone variables, the control run (Scenario X) simulates the level of SWC consistently too high, albeit the dynamics match observations fairly well (Fig. 3a). The model fails

to capture the overall dynamics of q including low and high flow events (Fig. 3b). For certain years, 2010 and 2011, snow periods are not simulated well.

As the turbulent fluxes have a distinct diurnal variation, we compare simulations and observations in Fig. 3 for four individual months. For the control run (Scenario X) the daytime (7am-7pm) LE values are slightly overestimated in June (Fig. 4a), while underestimated in all other months (Fig.

4b-d). For H both the daytime and nighttime values are overestimated in all four months (Fig. 4e-h). Thus, CLM5 highly overestimates H based on look-up parameter values and is not capable of simulating negative nocturnal H (Fig. 2d and 4e-h). In winter, the CLM5 control run simulates small negative H during night, but $H_{obs}$ is much lower than $H_{sim}$ (Fig. 4g).

### 3.2   Analysis of multi-objective calibration results

As expected, calibration enhances CLM5's ability to simulate the dynamics of the energy fluxes, recharge and soil moisture, though with a consistent overestimation of H (Fig. 1).

LE and H are linked through the energy balance and the partitioning of incoming energy into LE and H. In most calibrated scenarios, optimization against either one of the turbulent fluxes improve the other as well. Thus, the inverse calibration improves the simulation of both LE and H. When

comparing the initial model run (Scenario Z) with the calibrated scenarios in general, $H_{sim}$ and $H_{obs}$ match better in all the calibrated scenarios (Scenario A-K) than in the control run (Fig. 1). This applies to most metric types, but most evident for $\phi_H$, which is less than 100 in all scenarios (except Scenario I). This is the case regardless of whether H is used as calibration target (Scenarios E-H, J and K) or not (Scenarios A-D). In the same way as for H, Fig. 1 shows that summary statistics for

LE are likewise improved for all scenarios when comparing to the initial model run.

Scenarios A and D are, as expected, best in capturing the reduction in LE at harvest and grain-filling period of July and August. However, it is important to keep in mind that Fig. 2 shows daily mean over a six years period, and the variation in the timing of the harvest/grain-fill will affect the visual comparison in Fig. 2.

Figures 2c and d present results for the first week in June. When LE is used as target variable (Scenario A), H is overestimated, and vice versa when using H as target variable (Scenario E). The excess energy is placed on the other turbulent flux or on G (Fig.2).



Despite the improvement in both $LE_{sim}$ and $H_{sim}$, a clear discrepancy between $H_{sim}$ and $H_{obs}$ is found after calibration (Fig. 1), also for the single-objective optimization (Scenario E), where a bias of $ME_H = -11$ W m$^{-2}$ is found. $ME_H$ is negative in all scenarios with a value between -9 W m$^{-2}$ and -14 W m$^{-2}$. This is a very high absolute value especially when comparing to the mean value of the observations ($\mu_{Hobs} = 7$ W m$^{-2}$) (Fig. 1). The bias of $H_{sim}$ can also been seen on Fig. 2b, where the calibrated scenarios are not able to match mean daily $H_{obs}$ and the simulated values are higher than observations for most of the year. The same discrepancies between simulations and observations can be seen in Fig. 4e-h, where hourly $H_{obs}$ values are less than $H_{sim}$ values for all scenarios and for all months, especially at night. We see from Fig. 2d that CLM5 overestimates the nighttime negative H values.

For scenarios A and D, $LE_{sim}$ matches $LE_{obs}$ nearly perfectly in June (Fig. 4b and 2), while during the remaining seasons (Fig. 4a, 4c and 4d), $LE_{sim}$ is underestimated. There is only a slight difference in turbulent fluxes between Scenario A and D (Fig. 1, 2 and 4), thus including hydrological observations in the objective function does not have much effect on the results.

As expected the single-criteria optimization of LE (Scenario A) leads to the best summary statistics for LE (Fig. 1), but for H the best summary statistics are surprisingly obtained in scenario F and not in scenario E. In the same way, optimization against LE and q (Scenario B) gives better summary statistics for q than the single-objective optimization of q ($\phi_q = 58$ for Scenario B and $\phi_q = 81$ for Scenario I), and is capable of matching observed and simulated q to a better degree than other scenarios. However, in general the dynamics of q are not well simulated in any of the scenarios as reflected in $NSE_q$ being less than 0.46 for all scenarios.

The model is overall better in simulating the dynamics of LE as compared to H and the hydrological observations as evidenced by NSE for the different scenarios. This is also the case if LE is not included in the objective function. In all cases (except Scenario G), $NSE_{LE}$ is higher than $NSE_q$, $NSE_{SWC}$ and $NSE_H$ (Fig. 1).

The results demonstrate that it is important to include several datatypes in the optimization. Single-objective optimization against LE or H, respectively, leads to good results for the respective fluxes but deteriorates the simulation of the internal hydrological processes, especially SWC. The absolute level of simulated $SWC_{sim}$ is too high in the control run (Scenario X), but becomes much better when using site specific parameter values in the initial model run (Scenario Z) (not shown).

The information content of the different observation data types can be examined by comparing the model results of the different scenarios. When evaluating the model performance of Scenarios A to D, it is evident that when including q in the objective function (Scenario B) improves the fit of q





($\phi_q$ = 103 for Scenario A and $\phi_q$ = 58 for Scenario B) and SWC ($\phi_{swc}$ = 1301 for Scenario A and $\phi_{swc}$ = 341 for Scenario B), while still maintaining strong agreement with LE observations ($\phi$= 52 for Scenario A and $\phi_{LE}$ = 54 for Scenario B). On the other hand by including SWC in the objective function (Scenario C) actually also improves q ($\phi_q$= 103 for Scenario A and $\phi_q$ = 92 for Scenario C), while the match with LE observations becomes worse ($\phi_{LE}$ = 52 for Scenario A to $\phi_{LE}$ = 63 for Scenario C). When including both q and SWC in the calibration a good fit of LE and SWC and also an acceptable agreement with q observations can be obtained (Scenario D). Scenario D leads to the best overall model results. By including SWC in the parameter optimization leads to a good match between $SWC_{obs}$ and $SWC_{sim}$ (Fig. 3b).

Surprisingly, summary statistics (Fig. 1) do not change much when calibrating the dynamics of LE and H at the same time (Scenarios J and K). H is simulated with low accuracy independently whether LE is included in the objective function or not, while LE is simulated slightly worse in Scenario J than in scenario A ($\phi_H$ = 52 for Scenario A and $\phi_H$ = 61 for Scenario J). Including all four data types in the optimization (Scenario K) still leads to a bias of H simulations.

==FIGURE 5==

The parameter response space of CLM5 is complex, and the impacts of the parameters estimated on water and energy fluxes vary with different parameter value combinations. In general, Scenario D gives the best results. Figure 5 shows the identifiability, the relative error variance reduction and the estimates of 30 parameter estimated for Scenario D. The total height of each bar in Fig. 5a is

the identifiability of the pertinent parameter. The color-coding of each bar corresponds to the contribution by different eigencomponents spanning the calibration solution space to the identifiability. Warmer colors (red-yellow) correspond to singular values of smaller index (singular value of higher magnitude) and indicate that the parameter is less prone to measurement noise and more informed by observation data (Doherty, 2015).

The boundary between solution and null subspaces for Scenario D was set to 20. The 30 parameters show a broad range of identifiabilities, and if choosing a somewhat arbitrary qualitative identifiability level of 0.7 to mark cut-off between identifiable and non-identifiable parameters, then 14 out of 30 parameters are identifiable on the basis of the hourly observations of Rn, Sout, LE, q and SWC. The 14 identifiable parameters are primarily sand and clay fractions, LAI in

summer, height-top, medlyn and rootprof.

The parameters with the highest identifiability and which are mostly informed by data (warmer colors at Fig. 5) also have the highest relative error variance reduction. Hence, the information contained in the observation dataset constrain the identifiable parameters, while the non-



identifiable parameters are to a stronger degree constrained by expert-knowledge in the form of preferred-values in the Tikhonov regularization. Parameter confidence intervals reduces mostly for the parameters mostly informed by data.

FIGURE 6

Figure 6 shows the optimized parameter values, i.e. a) soil parameters, b) LAI and c) optical parameters, respectively. The optimized values for the plant functional type (PFT) parameters can be found in Table 1. Additionally, Fig. 6 indicates how much the parameters have moved from the a-priori loop-up table values and the initial parameter values.

The a-priori value for the PFT parameters are retrieved from global datasets, while the soil and
vegetation phenology parameters are linked to the study site location (Herbst et al., 2011; Vasquez, 2013). Sand1 and clay1 determine the hydraulic properties of the root zone. Sand1 is highly informed by data (warmer colors in identifiability plots, Fig. 5a), which is also seen from the narrow post-calibration confidence interval (Fig. 5b). According to the local information (Vasquez, 2013), the soil at the field site is sandy with only very little clay content. Most calibrated scenarios
obtain reasonable soil texture values, where the sand content mostly varies between the look-up table value of 60% and up to 100%, and the clay contend is below 20%. Scenarios B and I obtain an unrealistic low sand fraction in the root zone and Scenario A obtain an unrealistic high sand fraction in the root zone (Fig. 6). All scenarios which include q in the objective function reduce the fraction of sand in the soil layer below root zone (sand2). We know that this is incorrect and that
the soil texture becomes coarser with depth (Haarder et al., 2015). All scenarios not including q in the objective function have an expected high sand content below the root zone (Fig. 6a). Scenario A on the other hand has an unrealistic high value for the sand fraction of sand2. In generally the clay content is much less informed by data than sand (Fig. 6a)

The values of PFT parameters are listed in Table 1. All a-priori values of PFT parameters (except
medlyn) are nearly identical for the different vegetation types in the look-up tables of CLM5 (Lawrence et al., 2019a). Thus, specification of individual initial parameter values for each PFT is not possible.

Medlyn is a parameter of the stomatal conductance model. The parameter determines the degree of stomatal opening and has a critical impact on the stomatal responses in the soil-root-stem-leaf
system. The optimal value for medlyn varies between 3.38 and 5.75 (Table 1).

Rootprof is the root distribution parameter that determines the root fraction in each soil layer, and is critical for SWC of the soil. Roofprof is well informed by data and the regularization strategy allows the parameter value to move away from the initial value.





LAI shows similar patterns for all scenarios (Fig. 6b). As LAI parameters are non-identifiable in cold months, the values do not deviate much from the preferred values. The optimized LAIs enhance energy partitioning of LE and H during the grain filling and harvest phase in July and August (Fig. 2). The calibrated models match LE and H during the harvest period in July and

5   August better than the control run.



## 4    Discussion

The results presented shows that multi-objective calibration enhances the ability of CLM5 to represent both energy and hydrological processes considerably. This result is expected to be applicable elsewhere, particularly for low-lying agricultural areas subject to high

evapotranspiration. In line with Gupta et al. (1999), it was also demonstrated that optimization using a single-criterion objective function is less suitable as the internal hydrological processes are not represented adequately. In contrast, multi-objective parameter estimation considerably enhances the ability of CLM5 to simulate observed energy and hydrology data. According to the summary statistics, Scenario D (calibrated against LE, q and SWC) gives the best overall

representation of all data types (Fig. 1). Compared to the control run (Scenario Z), Scenario D reduce RMSE with 27%, 2%, 9% and 31% for LE, H, q and SWC, respectively.

In the following, we will discuss issues with respect to the energy and hydrology representation of the model, the calibration approach and the parameter uncertainty. However, to begin with, we will elaborate on the issue of land surface energy balance closure with respect to calibration of a LSM.

Throughout the discussion, we will outline potential future work within the subject of the study.

### 4.1    Energy balance closure

The eddy covariance (EC) method is generally regarded as the best practical method for measuring turbulent energy fluxes at the land surface, however numerous studies have documented the lack of energy balance closure (Foken et al., 2006; Franssen et al., 2010; Stoy et al., 2013). As

measurements of Rn is generally trusted, an underestimation of the turbulent fluxes appears likely because the sum of the energy fluxes is less than Rn (Foken et al., 2011). The  observation data from the field site (Ringgaard et al., 2011) show that incoming available energy (Rn minus G) on average exceeds the turbulent energy fluxes (LE and H) by 21% and the data is thus subject to land surface energy imbalance (Denager et al., 2020). Since LSMs conserve energy, the conclusions

from LSM calibration studies using turbulent fluxes as target variables, rest on the premise of closure of the observed energy fluxes. This is in contradiction and therefore Scenarios J and K are fundamentally incorrect as it is not possible to match $LE_{obs}$ and $H_{obs}$, simultaneously.

CLM5 simulates LE and H explicitly based on physical laws. Nonetheless, the regularization approach used in this study fails to identify parameter values to match uncorrected $H_{obs}$ with $H_{sim}$

(Scenario E). It is especially challenging to match negative H during winter and nocturnal periods, where the overlying air is warmer than the surface and sensible heat is therefore transported downwards (Fig. 2d and 4). There may be structural limitations of CLM5 that prevent a good match to H. However, as the observed incoming and outgoing energy is imbalanced (Denager et



al., 2020) and the model maintains Rn (Eq. 1), there is excess energy in the model, which CLM5 transmits to H and G (results not shown). G is often considered as a residual term for closing the energy balance in CLM5 e.g. Kracher et al. (2009). Denager et al. (2020) concluded by comparison to water balance measurements that the imbalance of the EC method at the specific field site is to a

less degree caused by errors in the LE estimates, but can mainly be attributed to errors in the other energy flux components or unaccounted effects.

Contrary to this study, many studies have tested LSMs using corrected flux observations of H and LE that fulfill energy closure (Carrillo-Rojas et al., 2020; Davison et al., 2016; Dombrowski et al., 2022; Larsen et al., 2016; Pauwels and De Lannoy, 2011). A few studies have tested LSMs using

both corrected and uncorrected turbulent fluxes (Chen et al., 2018), while some studies do not indicate whether turbulent energy fluxes are corrected or not (De Lannoy et al., 2011; Göhler et al., 2013; Hou et al., 2012). Chen et al. (2018) applied both corrected and uncorrected LE and H from FLUXNET for testing a point-scale CLM4.5 over open sites, and found that simulations matched uncorrected LE better than corrected LE, and, as energy-balance correction methods increase the

LE values, they found that CLM4.5 underestimated FLUXNET corrected LE.

## 4.2  Worth of observation data

Physically, LE depends on both energy flux and water avaliability. Aside from LE, moisture information is clearly central for optimizing the internal hydrological processes of CLM5. Other studies have also shown the appropriateness of SWC in optimizing the hydrological state in LSMs

(De Lannoy et al., 2011; Zhang et al., 2017). Similar to LE, groundwater recharge, q, also describe the water exchange, however as long as LE data is available, q data only gives minor additional information to the calibration.

Data uncertainty has been discussed in Denager et al. (2020) and we are generally confident with the accuracy of our forcing and hydrological data. To improve the simulation of soil water flow in

LSMs, we followed the suggestion by Rosero et al. (2010) and used percolation observations in the parameter optimization process. To capture the diurnal dynamics of energy and water fluxes the optimization is based on hourly time steps. However, given the design of the lysimeters at the field site where recharge water is collected at a sloping face at the bottom of the lysimeters there may be a temporal mismatch between model simulations and observations. Through each of the four

lysimeters have a surface area of 3.2 x 3.88 m, their total area is much smaller than the footprint of the EC system.



### 4.3 Calibration approach

As six years of observations are available for all major water and energy balance components at the field site, there is a potential for studying the long-term effects on the seasonal energy and water fluxes and variables. However, the target of the applied calibration approach is the dynamics of the
24-hour cycle of hourly observations rather than the seasonal energy and water balance components.

Sun et al. (2013) found that parameter optimization by PEST only led to small improvements in performance of CLM4.0. In the present study, we were able to obtain considerable improvements by parameter optimization using singular value decomposition and Tikhonov regularization
implemented in the PEST software package. This approach is more computational effective than general Bayesian approaches that require a large number of model simulations to estimate parameter and predictive uncertainty such as the stochastic Markov-chain Monte Carlo inversion of CLM4 presented by Sun et al. (2013). Another approach was presented by Zhang et al. (2017) who evaluated different data assimilation methods for soil texture parameter estimation in CLM.

### 4.4 Evaluation of optimal parameters values

Some CLM5 parameters, e.g. LAI and height_top, are physically meaningful and can be inferred directly from observations, while other parameters, e.g. displar, dleaf, medlyn, rootprof and z0mr can be viewed as conceptual representations for which useful values cannot be directly measured.

Aside from the stomatal resistance, LAI also directly controls actual evapotranspiration, and as the
sum of LE and H is constrained by the energy preservation in CLM5, LAI consequently determines both LE and H.

Theoretically, LAIs should not change between calibration scenarios, and most scenarios actually show very similar LAI and SAI values. Scenarios A-D show well-constrained $LAI_{jun}$ values between 4.14 and 5.37. We did not consider SAI parameters as adjustable parameters, but
preliminary model calibrations including SAI showed that the decrease of $LAI_{jul}$ and $LAI_{aug}$ were compensated in nearly all scenarios by an increase in $SAI_{jul}$ and $SAI_{aug}$. However, we do not expect SAI to have considerable influence on turbulent fluxes and hydrological variables. The increase in LAI in some scenarios in September probably reflects the emerging of cover crop.

When CLM5 is run in satellite phenology mode, it is not capable of simulating the year-to-year
variation in germination, leaf emergence, harvest etc., as all years are assumed to follow the same pattern. The energy partitioning in July and August is simulated better some years than others, but despite the alignment of distinct yearly phenology in CLM5, the abrupt decrease in LE (averaged over 6 years) at grain filling/harvest, is quite well simulated (Fig. 2a). Calibration of CLM5 with



inclusion of the biogeochemistry (BGC) model is beyond the scope to this paper, but as CLM5-BGC applies carbon and nitrogen cycle functionality, the CLM5-BCG replaces phenology with prognostic variables. These variables change dynamically with meteorological forcing, soil moisture and nutrient availability (Cheng et al., 2021). Inclusion of the BGC-module in CLM5

would further enable simulations of cover crops schemes (Boas et al., 2021). According to Boas et al., (2021), the cover crop scheme helped to match the observed energy balance.

It is a large disadvantage when calibrating LSMs that many important parameters are often hard coded (Davison et al., 2016). Adjusting those hard coded parameters requires manual alteration of the appropriate code lines and subsequent recompiling before every parameter trial in the

calibration routine. This limits the calibration process and the ability of the model to describe important processes (Mendoza et al., 2014).

The model uses pedo-transfer functions to estimate the soil hydraulic properties, which is a useful approach for large-scale applications. However, for local scale applications as in this study it would have been more appropriate to be able to specify the hydraulic properties directly. We observed

that CLM5 overestimates the recharge during spring and summer, indicating that the representation of the hydraulic properties are inadequate when estimated from pedo-transfer functions of optimized soil texture. A large number of the former studies regarding parameter estimation and parameter sensitivity in CLM5 relate their analysis to the hydrologic parameters (e.g. hydraulic conductivity) rather than evaluating the model parameters in the pedo-transfer functions (e.g.

percentage of sand and clay) (Göhler et al., 2013; Hou et al., 2012; Huang et al., 2013; Sun et al., 2013).

De Lannoy et al. (2011) analyzed the effect of different soil texture specifications on simulations of SWC, LE and H using CLM3.5 and concluded that the impact of soil texture on energy fluxes is minor but impact on water storages characteristics was significant. The present study found that the

soil texture parameters (especially in the root zone) are identifiable also in the single-objective calibration of Scenario A.

It should be noted that although soil texture is defined as a proportion of sand and clay, and therefore has the unit of percentage, individual values of sand or clay >100% is conceivable in CLM5, because the parameter interval were set >100%. In some scenarios, we obtain a sum of sand

and clay slightly above 100% but is not considered an critical issue as the textural percentages only enter as parameters in the pedo-transfer function for the hydraulic properties.

Similar to other sensitivity studies of CLM, we find that the stomatal conductance parameter (medlyn) and teh soil parameters are highly significant (Göhler et al., 2013). In contrast, Hou et al.,



(2012) and Huang et al. (2013) found that in CLM4 subsurface generation parameters (distribution of surface runoff with depth, max subsurface drainage and specific yield) are the most important parameters for LE, H and runoff, while soil texture parameters (Clapp and Hornberger parameter b and porosity) are of secondary significance. However, which parameters that are most sensitive can

vary from site to site and from season to season, and the significance of parameters also depend on which target variable that is considered. As our cropland field site has a shallow root zone, the unsaturated zone parameters (e.g. soil texture in top layer) become more important.

The a-priori value of 0.943 for *rootprof* is similar for all grass and crop PFTs (Lawrence et al., 2019a). Thereby off-the-shelf CLM5 does not distinguish root density for different types of grasses

and crops. There is a clear possibility to constrain individual *rootprof* parameter values for different land-cover types. We found the *rootprof* parameter to be highly identifiable and thereby highly informed by observation data of LE, q and SWC. Our optimized values of the parameter *rootprof* for the scenarios including SWC in the objective function (Scenario B and C), are substantial different from the a-priori value (*rootprof* = 0.39 (Scenario B), *rootprof* = 0.56 (Scenario C)).

However, the optimized values of rootprof seem reasonable, as they imply an increase of the root density near surface and a reduction at deeper soil layers, which fit well with the spring and winter barley cultivated at the agricultural field.



## 5   Conclusion

In this study, we explore how parameter estimation techniques can be used for improving the hydrological processes in a state-of-the-art LSM. The results indicate that mathematical
regularization is a compelling method to improve the current practice of using look-up tables to define parameter values in LSMs.

Through a case-study of an agricultural field in western Denmark with six years of extensive observations, we demonstrate that calibrating a point-scale CLM5 using i) multi-objective calibration, ii) truncated singular value decomposition and iii) Tikhonov regularization using
combinations of hourly time series of latent heat, sensible heat, soil moisture and groundwater recharge from 2010-2015, can considerably improve the characterization of the energy and water fluxes.

We found that parameter optimization of CLM5 using soil moisture data enhanced the ability of the model to describe the temporal patterns of moisture storage within the root zone. Calibration also
considerably improved the energy partitioning of LE and H during the summer period, and revealed good reproduction of observed and simulated LE and H during the grain filling and harvest period in July and August.

However, we found that H was biased the rest of the year as the simulated H was clearly overestimated. It was not possible to fine tune parameters to match observed H, which suggests that
observed H needs to be corrected to match simulations.

Additionally, we evaluated the post-calibration uncertainties of the model parameters through the two statistics of identifiability and relative error variance reduction. Identifiability indicates to which extent the parameter is informed by observation data. Using LE, q and SWC as target variables, we found that the identifiable parameters were soil texture, monthly LAI in summer, the
stomata conductance model parameter (medlyn) and the root distribution parameter (rootprof).

Our results underline the necessity of parameter calibration using available observations of energy and hydrological fluxes to obtain an optimal parameter set for CLM5. We anticipate that the results from this study contribute to improvements in model characterization of water and energy fluxes, especially when EC flux data are available.



## 6    Data availability

The consistently forcing and calibration data from the site are available at

https://www.enoha.eu/network/hobe

## 7    Author contribution

TD, TSO, MCL and KHJ designed the study. TD did all numerical modelling, analyses, and
designed the figures. TD took the lead in writing the manuscript. All authors discussed results and
provided critical feedback to the manuscript drafts.

## 8    Declaration of interests

The authors declare that they have no known competing financial interests or personal relationships
that could have appeared to influence the work reported in this paper.

## 9    Acknowledgements

The Villum Foundation has funded the hydrological observatory, HOBE, and the research reported
in this paper. We are very thankful for the opportunities that this donation provides. Additionally
we would like to thank Theresa Boas and Lukas Strebel, Forschungszentrum Juelich, and Rena
Meyer, University of Oldenburg, for help and guidance.



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





Table 1: Look-up table, initial and optimized parameter values for all scenarios. LAI and optical parameter values can be found in Appendix. —ıı— means that the value is identical for Scenario X and Z.

| Scenario / Target variable | | Units | X (look-up table value) | Z (initial value) | A (LE) | B (LE and q) | C (LE and SWC) | D (LE, q and SWC) | E (H) | F (H and q) | G (H and SWC) | H (H, q and SWC) | I (q) | J (LE and H) | K (LE, H, q and SWC) |
|---|---|---|---|---|---|---|---|---|---|---|---|---|---|---|---|
| sand, root zone | sand1 | % | 60 | 90 | 53 | 59 | 86 | 99 | 86 | 102 | 82 | 75 | 54 | 86 | 78 |
| sand, below root zone | sand2 | % | 60 | 90 | 127 | 33 | 100 | 57 | 97 | 60 | 93 | 73 | 46 | 100 | 78 |
| clay, root zone | clay1 | % | 10 | 4 | 6 | 16 | 5 | 14 | 4 | 4 | 4 | 4 | 6 | 4 | 5 |
| clay, below root zone | clay2 | % | 10 | 4 | 5 | 22 | 5 | 14 | 4 | 8 | 4 | 5 | 9 | 4 | 5 |
| canopy top height | height top | m | 0.5 | —ıı— | 0.63 | 0.95 | 0.54 | 1.04 | 0.38 | 0.22 | 0.52 | 0.42 | 0.69 | 0.44 | 0.39 |
| displacement height to canopy top height | displar | - | 0.68 | —ıı— | 0.86 | 0.79 | 0.92 | 0.62 | 0.7 | 0.91 | 0.72 | 0.73 | 0.7 | 0.73 | 0.71 |
| Characteristic leaf dimension | dleaf | m | 0.04 | —ıı— | 0.04 | 0.04 | 0.04 | 0.03 | 0.04 | 0.03 | 0.04 | 0.04 | 0.03 | 0.04 | 0.04 |
| Ratio of momentum roughness length to canopy top height | z0mr | - | 0.12 | —ıı— | 0.1 | 0.07 | 0.11 | 0.08 | 0.1 | 0.09 | 0.12 | 0.1 | 0.15 | 0.07 | 0.11 |
| Leaf resistance parameter | medlyn | - | 5.79 | —ıı— | 3.96 | 5.34 | 3.88 | 4.71 | 5.17 | 5.75 | 4.99 | 4.38 | 5.6 | 4.28 | 3.95 |
| root distribution parameter | root-prof | - | 0.94 | —ıı— | 0.78 | 0.39 | 0.92 | 0.56 | 0.82 | 0.76 | 0.95 | 0.93 | 0.99 | 0.81 | 0.9 |





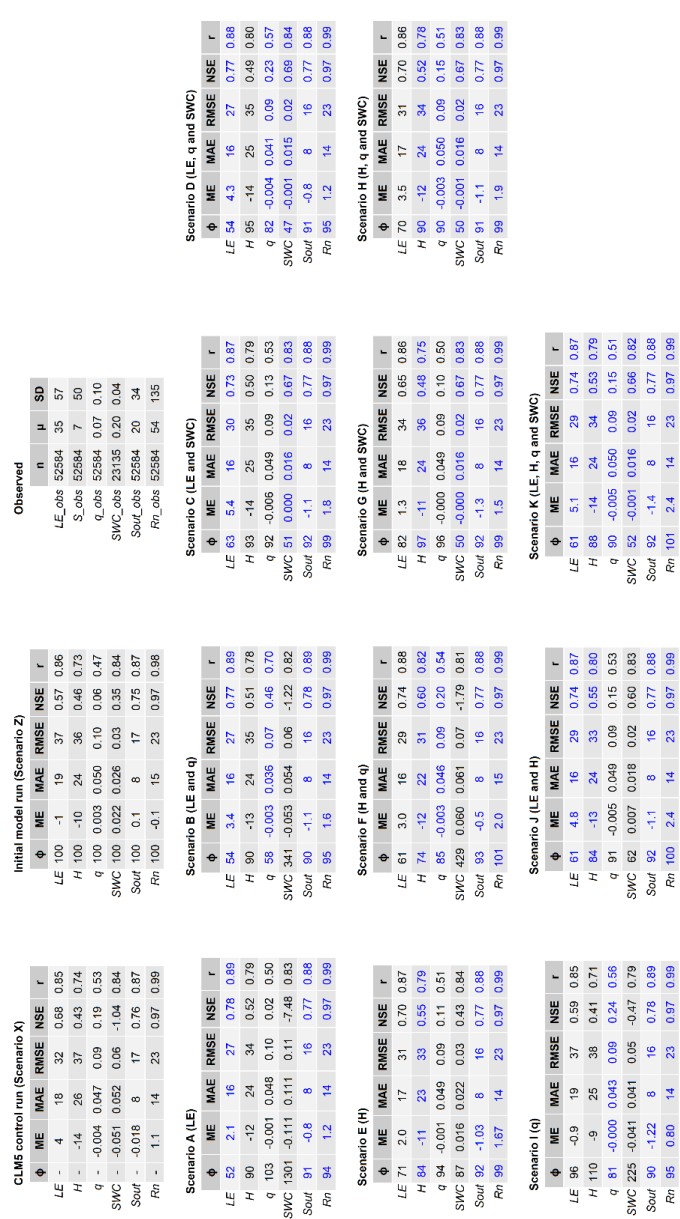

*Figure 1: Summary statistics. units: NSE and r are unit less, ME and MAE for H and LE are [W m⁻²], ME and MAE for q is [mm/h] and ME and MAE for SWC is [m³m⁻³]. φ and RMSE for H and LE is [(W m⁻²)²], φ and RMSE for q is [(mm h⁻¹)²] and φ and RMSE for SWC is [(m³ m⁻³)²]. Blue color indicate that in variable were included in the calibration for the given scenario*



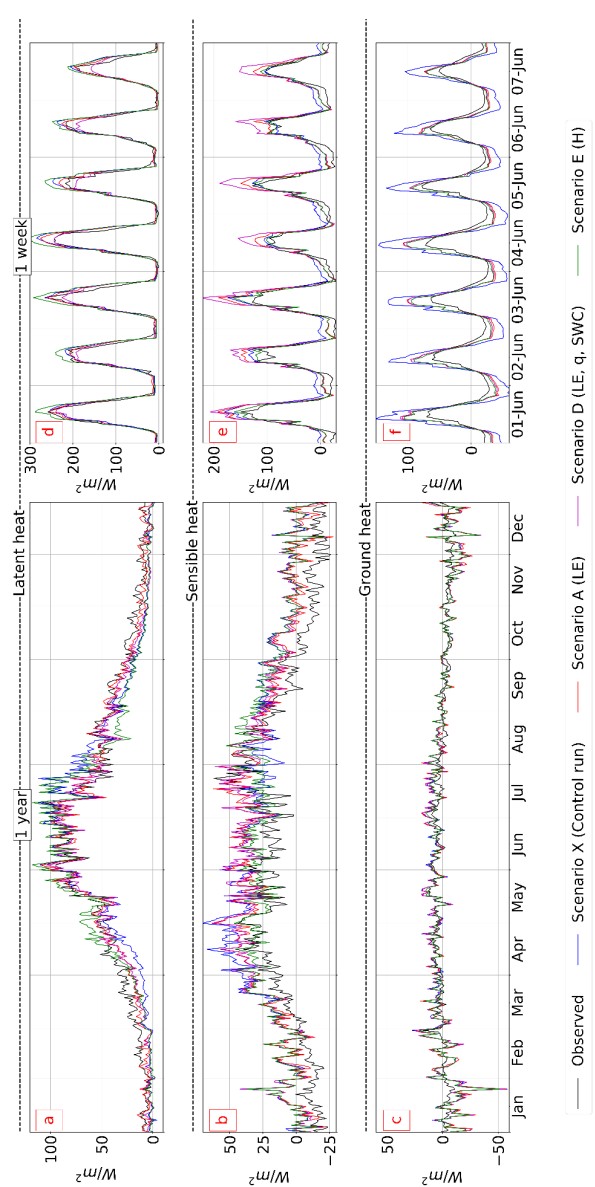

*Figure 2: Observed and simulated LE and H (daily mean 2010-2015) for control run, Scenario A, D and E over a year (left), and (hourly mean 2010-2015) over a one week period in June (right).*





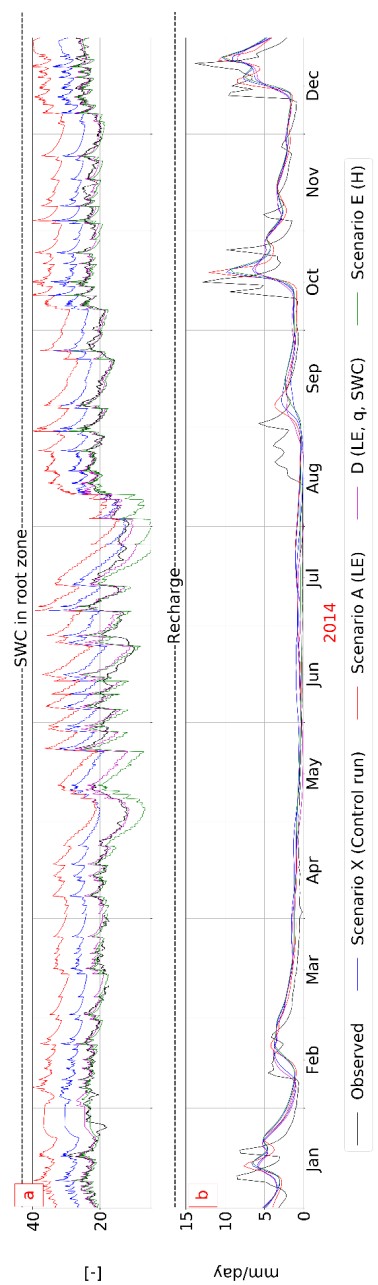

*Figure 3: Observed and simulated SWC and q for control run for Scenarios A, D and E in 2014.*



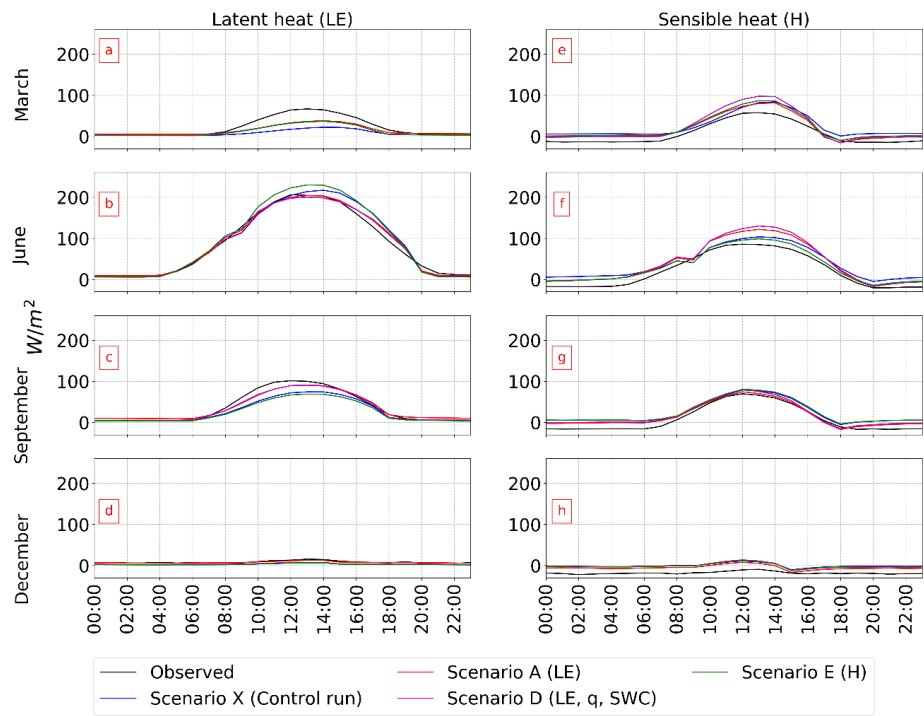

*Figure 4: Seasonal daily cycle of observed and simulated (hourly mean 2010-2015) LE and H.*





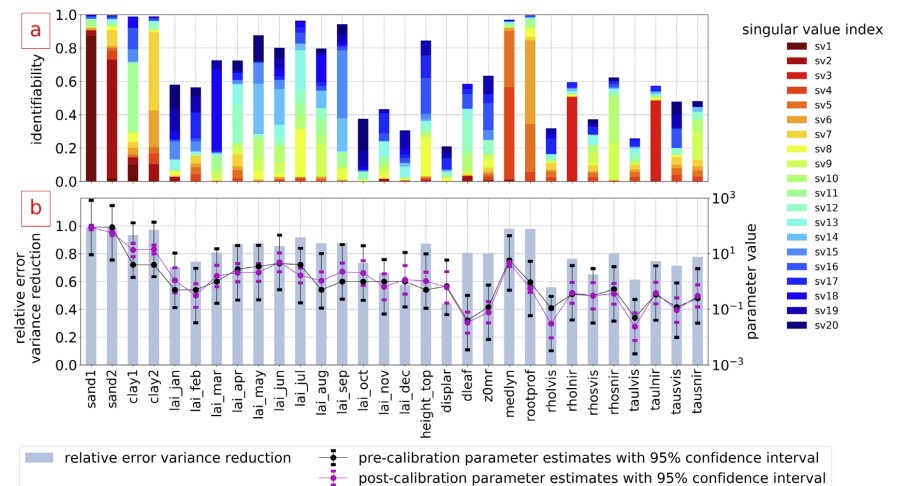

*Figure 5: Identifiability (subplot a), relative error variance reduction and optimized parameter values (subplot b) for Scenario D. The total height of the bars in subplot a) indicates identifiability of each parameter and the color-coding of each bar corresponds to the contribution of the singular values to the identifiability. Please note the logarithmic scale on the secondary y-axis of subplot b).*





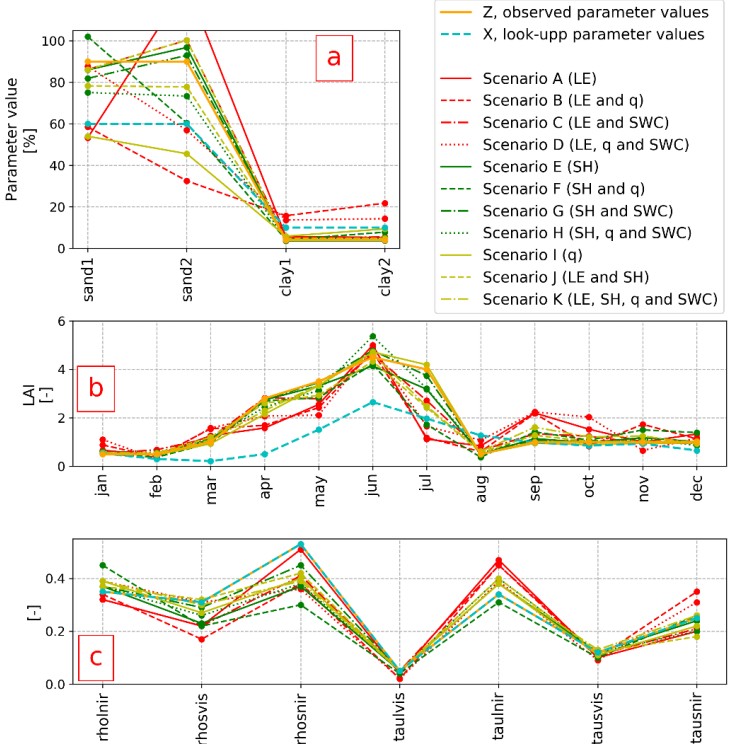

*Figure 6: Optimized parameter values for all scenarios. a) soil parameters, b) LAI and c) optical parameters.*





*Appendix A: Look-up table, initial and optimized LAI for all scenarios.*

| Scenario | Unit | X look-up table value | Z initial value | A LE | B LE and q | C LE and SWC | D LE, q and SWC | E H | F H and q | G H and SWC | H H, q and SWC | I q | J LE and H | K LE, H, q and SWC |
|---|---|---|---|---|---|---|---|---|---|---|---|---|---|---|
| Leaf Area Index | - | | | | | | | | | | | | | |
| Jan | - | 0.53 | 0.5 | 0.66 | 0.88 | 0.52 | 1.1 | 0.52 | 0.52 | 0.52 | 0.6 | 0.56 | 0.54 | 0.56 |
| Feb | - | 0.3 | 0.5 | 0.45 | 0.38 | 0.68 | 0.31 | 0.48 | 0.37 | 0.53 | 0.57 | 0.46 | 0.54 | 0.54 |
| Mar | - | 0.21 | 1 | 1.26 | 1.55 | 1.17 | 1.58 | 1.11 | 0.96 | 1.01 | 1.13 | 0.93 | 1.2 | 1.08 |
| Apr | - | 0.5 | 2.8 | 1.58 | 1.69 | 2.82 | 2.07 | 2.74 | 2.7 | 2.75 | 2.44 | 2.15 | 2.6 | 2.3 |
| May | - | 1.51 | 3.5 | 2.55 | 2.42 | 2.79 | 2.11 | 3.32 | 2.83 | 3.46 | 3.13 | 3.33 | 2.93 | 3.36 |
| Jun | - | 2.65 | 4.5 | 5 | 4.73 | 4.63 | 4.9 | 4.14 | 4.21 | 4.78 | 5.37 | 4.72 | 4.33 | 4.64 |
| Jul | - | 1.97 | 4 | 1.12 | 1.18 | 2.71 | 1.65 | 3.18 | 1.72 | 3.74 | 3.21 | 4.19 | 2.42 | 2.48 |
| Aug | - | 1.27 | 0.5 | 0.84 | 0.6 | 0.65 | 1.07 | 0.5 | 0.37 | 0.51 | 0.55 | 0.5 | 0.58 | 0.61 |
| Sep | - | 0.98 | 1 | 2.2 | 2.18 | 1.34 | 2.24 | 1.14 | 1.42 | 1.06 | 1.11 | 0.95 | 1.28 | 1.61 |
| Oct | - | 0.85 | 1 | 1.53 | 0.83 | 1.24 | 2.03 | 0.97 | 1.08 | 1.05 | 0.95 | 0.89 | 1.02 | 1.21 |
| Nov | - | 0.93 | 1 | 1 | 1.73 | 0.95 | 0.65 | 1.18 | 1.5 | 1.03 | 1.12 | 0.95 | 1.02 | 1.25 |
| Dec | - | 0.65 | 1 | 1.33 | 1.13 | 0.95 | 1.16 | 1.01 | 1.39 | 0.97 | 0.86 | 0.96 | 1.03 | 0.93 |





Appendix B: Look-up table, initial and optimized optical parameters for all scenarios.

| Scenario | Target variable | Unit | X look-up table value | Z initial value | A LE | B LE and q | C LE and SWC | D LE, q and SWC | E H | F H and q | G H and SWC | H H, q and SWC | I q | J LE and H | K LE, H, q and SWC |
|---|---|---|---|---|---|---|---|---|---|---|---|---|---|---|---|
| leaf reflectance - shortwave | rholvis | - | 0.11 | —η— | 0.03 | 0.03 | 0.09 | 0.03 | 0.1 | 0.08 | 0.11 | 0.09 | 0.09 | 0.08 | 0.09 |
| leaf reflectance – longwave | rholnir | - | 0.35 | —η— | 0.32 | 0.34 | 0.32 | 0.39 | 0.37 | 0.45 | 0.37 | 0.37 | 0.39 | 0.37 | 0.39 |
| stem reflectance - shortwave | rhosvis | - | 0.31 | —η— | 0.22 | 0.17 | 0.22 | 0.31 | 0.23 | 0.22 | 0.29 | 0.26 | 0.27 | 0.32 | 0.3 |
| stem reflectance – longwave | rhosnir | - | 0.53 | —η— | 0.51 | 0.38 | 0.41 | 0.36 | 0.37 | 0.3 | 0.45 | 0.38 | 0.4 | 0.42 | 0.39 |
| leaf transmittance - shortwave | taulvis | - | 0.05 | —η— | 0.04 | 0.02 | 0.05 | 0.02 | 0.05 | 0.04 | 0.05 | 0.05 | 0.05 | 0.05 | 0.05 |
| leaf transmittance – longwave | taulnir | - | 0.34 | —η— | 0.47 | 0.45 | 0.45 | 0.39 | 0.38 | 0.31 | 0.38 | 0.4 | 0.38 | 0.4 | 0.38 |
| stem transmittance - shortwave | tausvis | - | 0.12 | —η— | 0.1 | 0.1 | 0.11 | 0.09 | 0.12 | 0.1 | 0.12 | 0.12 | 0.11 | 0.12 | 0.13 |
| stem transmittance – longwave | tausnir | - | 0.25 | —η— | 0.2 | 0.35 | 0.21 | 0.31 | 0.24 | 0.26 | 0.25 | 0.2 | 0.22 | 0.18 | 0.26 |