# Peer review of "Point-scale multi-objective calibration of the Community Land Model Version 5.0 using in-situ observations of water and energy fluxes and variables"

_EGUsphere, 2022_

## Author Comment (AC1)

**Reviewer specific comments:**
Recent literature on spatial calibration of LSM is missing from Introduction section.
**Reply:**
We agree that recent literature on spatial calibration of LSMs is missing in the Introduction section. However, there are not many studies where land surface model parameters are estimated. We already write that "… only a limited number of studies have dealt with calibration and sensitivity analysis of the energy and hydrology parameters in LSMs" (p3, line 15-16). Because we apply CLM in point-scale mode, the Introduction section put emphasis on the point-scale LSM calibration, however we agree that especially (Demirel et al., 2018; Mendiguren et al., 2017) is relevant, and we will include those in the Introduction section.

We will include the suggested papers as references:

(Tangdamrongsub et al., 2017) at page 2 line24

(Demirel et al., 2018) at page 2 line 25

(Mendiguren et al., 2017) at page 3 line 24

(Lane et al., 2021) at page 2 line 28

**Reviewer:**
-Page 7 Line 11: "1000 years" please explain how?

**Reply:**
1000 years is a somewhat arbitrary number to ensure that the model had reach a quasi-equilibrium. According to "CLM5 userguide" CLM5.0-BGC-Crop needs at least 1000 years of spin-up. Our CLM model does not include ecosystem carbon, however is takes approximately 150 years of spinup for soil temperature to reach equilibrium.

**Reviewer:**
-P7L14: "final simulations" why only before final simulations and not during calibration? Please explain more..
**Reply:**
We included four years of spin-up preceding each and every simulation in the calibration. We see that this is not clearly written, and we will rewrite the sentence.

**Reviewer:**
-P8L5 to11: this paragraph should be moved to the section 2.3 describing calibration approach to avoid repetition.
**Reply:**
Ok, we will do that.

**Reviewer:**
-P8L27: "Focus was given to a set of 30 time-invariant model parameters." Apparently no sensitivity analysis was applied? Why?
**Reply:**
John Doherty (personal communication) recommends Highly Parameterized Inversion, were most parameters are included in the calibration. The regularization approach will keep the insensitive parameters at their preferred values. We are aware that the calibration time could be reduced if removing some parameters from the calibration,

however we were interested in studying how much the parameter values deviated from the look-up table values after regularization.

**Reviewer:**
-In a calibration framework it is essential to apply SA first to reduce search dimension. May be some of the 30 parameters have zero influence on the objective function? Did you utilize PEST's local sensitivity analysis option?
**Reply:**

We did do a local sensitivity analysis with PEST (see figure below). However as explained in the former question we decided to include all parameters in the calibration.

[Figure]

*Figure 1: Local sensitivity analysis from PEST. The analysis has been done on initial parameter value of LAI in March, September, October, November and December of 0.5 instead of 1, because log-transformation of those parameters would reveal a sensitivity of zero.*

**Reviewer:**
-Section 2.3: The reader can be curious about several details of the calibration framework.
**Reply:**
We agree that readers could be curious about the details of the calibration framework. We have answered your questions in the following and we will incorporate the information on the calibration in the manuscript.

1)what was the user defined maximum number of iterations for such a sophisticated mode?
**Reply:**
Yes, maximum number of iterations were defined as 50 iterations. The only scenario reaching this were scenario F.

**Reviewer:**
2)computer runtime statistics and cluster properties (logical processors, ram capacity, intel/amd etc)
**Reply:**
One model run took about 10 minutes on a Linux server (Intel Xeon Gold 6148 processor, 20 cores, 380 GB RAM).

**Reviewer:**

2)Pest has three search algorithms LM, SCE-UA and CMAES. Can "Tikhonov regularization" be used together with one of these search algorithms?
**Reply:**
We chose to apply the gradient-based nonlinear Gauss-Marquardt-Levenberg method implemented in PEST, were the calculation of finite–difference derivatives are used in the inversion process. We did that because those often use fewer models runs that alternative optimization techniques (Doherty, 2015). We introduced Tikhonov regularization to honor the observed parameters values as prior knowledge. If using the global optimizer of CMAES_P (also implemented in the PEST suite) we could have used "pseudo-regularization" (Doherty, 2018) were credence to parameters derivatives is done by weight adjustments.

**Reviewer:**
3) sharing PEST control file ".pst" in appendices (or supplementary) can be good for this open access journal.
**Reply:**
Ok, we will share the .pst file either as appendix or supplementary.

**Reviewer:**
-only eq 10 is bias insensitive metric. Why the authors did not choose a spatial metric focusing on patterns of fluxes in growing season? Evaluating hourly (unstable) fluxes can be misleading. Instead evaluating monthly patterns of SWC, AET, SM can be a robust guide for the model. Fig 2-3-4 are showing only temporal aspects of the fluxes/states but this kind of finite element based LSMs can provide maps outputs. The authors should show also some map results. Looking at only time series can be boring.
**Reply:**
As we apply CLM in point-scale mode it is not possible to include a spatial metric or showing the results at maps.
We write that: "The target of the applied calibration approach is the dynamics of the 24-hour cycle of hourly observations rather than the seasonal energy and water balance components." (page 21, line 4) Therefore, we did not include the monthly and seasonal patterns.

**Reviewer:**
-why Pareto approach was not used for multi-objective calibration to avoid dominating solutions. Pareto DDS algorithm (available in Ostrich) could offer multiple non dominating solutions. PEST doesn't include this algorithm yet.
**Reply:**
PEST is one of the most well-developed inversion and parameter uncertainty software programs. However, we agree that it would have been an opportunity to use the Pareto DDS algorithm (in Ostrich). Obtaining multiple non dominating solutions would show if the look-up table parameters values were inside the parameter values interval of those solutions. However, the single parameters value set of minimum error variance obtained in our approach, somehow correspond to the method of using a single-set look-up table values as generally done in LSMs, so that those two parameter set can be directly compared.

**Literature:**

Demirel, M. C., Mai, J., Mendiguren, G., Koch, J., Samaniego, L., & Stisen, S. (2018). Combining satellite data and appropriate objective functions for improved spatial pattern performance of a distributed hydrologic model. *Hydrology and Earth System Sciences*, *22*(2), 1299–1315. https://doi.org/10.5194/hess-22-1299-2018
Doherty, J. (2015). *Calibration and Uncertainty Analysis for Complex Environmental Models - The PEST book*.

Doherty, J. (2018). *PEST - Model-Independent Parameter Estimation - User manual Part I*.

Lane, R. A., Freer, J. E., Coxon, G., & Wagener, T. (2021). Incorporating Uncertainty Into Multiscale Parameter Regionalization to Evaluate the Performance of Nationally Consistent Parameter Fields for a Hydrological Model. *Water Resources Research*, *57*(10), 1–19. https://doi.org/10.1029/2020WR028393

Mendiguren, G., Koch, J., & Stisen, S. (2017). Spatial pattern evaluation of a calibrated national hydrological model - A remote-sensing-based diagnostic approach. *Hydrology and Earth System Sciences*, *21*(12), 5987–6005. https://doi.org/10.5194/hess-21-5987-2017

Tangdamrongsub, N., Steele-Dunne, S. C., Gunter, B. C., Ditmar, P. G., Sutanudjaja, E. H., Sun, Y., Xia, T., & Wang, Z. (2017). Improving estimates of water resources in a semi-arid region by assimilating GRACE data into the PCR-GLOBWB hydrological model. *Hydrology and Earth System Sciences*, *21*(4), 2053–2074. https://doi.org/10.5194/hess-21-2053-2017

---

## Author Comment (AC2)

**Reviewer2 (R2) specific comments (C):**

**R2_C1:**
Denager et al. implemented multi-objective calibration of point-scale CLM5 using several types of flux/states observations of LE, H, recharge (q) and SWC from the Danish hydrological observatory HOBE. This topic of constraining model parameters against multi-source observations is quite relevant to the HESS journal, and it can be a valuable contribution to the community after addressing my following comments listed below. Additionally, the paper is clearly written and well-referenced; some parts require revisions, as follows. English typos should be double-checked, and some textual suggestions are further provided at the end. In the figures, I often can't distinguish individual scenarios. Differences between individual calibration scenarios are not clearly depicted. Please, improve the readability of the figures.

**Reply:**
To improve the readability, Figure 2, 3 and 4 will only include observed, scenario X and scenario D. Scenario A and E will be removed from the figures. Figure 1 will be changed to a table. See reply on R2_C6.

**R2_C2:**
1) Title requires modification. It needs to be clear from the title that the calibration is for one point-scale site.

**Reply:**
"Local-scale" will be added to the title such that it will read:

Local-scale multi-objective calibration of the Community Land Model Version 5.0 using in-situ observations of water and energy fluxes and variables

**R2_C3:**
2) The abstract should be more concise and to the point, highlighting concrete results of the present study, and quantifying the results. So, please remove/rewrite too general statements, which are probably better suited for discussion of the results or conclusions. E.g. I suggest removing "Furthermore, reliability of the optimized model parameters can be estimated by statistical measures such as identifiability and relative error variance reduction. As in most other eddy covariance studies, closure of the land surface energy balance is not achieved on observation data." The following statement, "The fact that CLM5 is not capable of matching sensible heat, not even with advanced parameter optimization of model parameter values, suggests that the lack of energy closure is due to biases in the sensible heat flux" is probably also not the most suitable one for the abstract. Instead, I would like to know from the abstract, which of the considered variable was most useful in improving the process representation. Did calibration of one variable improve the model's predictive skill of another (uncalibrated) variable? Which one? Also, an abstract should mention at which site (i.e., agricultural field observatory in Denmark) the CLM5 is established.

**Reply:**
The abstract will be rewritten according to the suggested improvements. The abstract will not focus on the energy balance but focus on calibration target variables and parameters. See suggested new abstract below:

**Abstract:** This study evaluates water and energy fluxes and variables in combination with parameter optimization of the state-of-the-art land surface model Community Land Model version 5 (CLM5), using six years of hourly observations of latent heat flux, sensible heat flux, groundwater recharge, soil moisture and soil temperature from an agricultural observatory in Denmark.
The results show that multi-objective calibration in combination with truncated singular value decomposition and Tikhonov regularization is a powerful method to improve the current practice of using look-up tables to define parameter values in land surface models. Using measurements of

turbulent fluxes as target variable, the parameter optimization is capable of matching simulations and observations of latent heat, especially during the summer period, while simulated sensible heat is clearly biased. Of the 30 parameters considered soil texture, monthly LAI in summer, stomata conductance and root distribution have the highest influence on the local-scale simulation results. The results from this study contribute to improvements of the model characterization of water and energy fluxes. The study underlines the importance of performing parameter calibration using observations of hydrologic and energy fluxes and variables to obtain optimal parameter values of a land surface model.

**R2_C4:**
3) Second half of the Introduction should clearly point out the research gap and your contribution to filling it in. Clearly stating the novelty of your manuscript somewhere in the last two paragraphs of the Intro.
**Reply:**
We will include the following paragraph in the introduction:

"In this study, we evaluate in-situ water and energy fluxes and variables at an agricultural field site in Denmark using the state-of-the-art LSM Community Land Model version 5 (CLM5) coupled to the optimization code PEST (Doherty, 2015). In most previous research, LSMs are not calibrated and instead use lookup tables to define parameter values. Here we identify values of important parameters in an LSM using multi-objective calibration in combination with regularization to improve the simulation of the hydrological processes."

Furthermore, we will improve the overall language of the introduction such that the novelty of the study stands out more clearly.

**R2_C5:**
4) Regarding the experimental design (Page 8, Line 8), please be consistent; earlier, you mention four variables (LE, H, recharge (q) and SWC); here, you mention six different observation data sources. Which one is correct, then? Please, synchronise, otherwise it is confusing. Table 1 already includes the calibrated parameter values, It is not clear how these parameters were identified when Table 1 was first introduced. From Table 1, it looks like you calibrated sand and clay contents directly. Was Clapp-Hornberger exponent B also part of the calibration process? As it is not part of Table 1. Please, clarify.
**Reply:**
Seven different observation data sources are used in the study, LH, H, q, SWC, Sout, Rn and (in the revised manuscript also) Tsoil. This will be stated clearly in the revised manuscript.
Clapp-Hornberger exponent B is not a calibration parameter. This will be clarified in the manuscript. The Clap-Hornberger B exponent is inherently defined in CLM5 from pedo-transfer functions of percentages of sand, clay and organic matter.

**R2_C6:**
5) Figure 1 is a rather set of tables than a figure. Increase the font and readability of the Table.
**Reply:** Thank you for the comment. We have discussed this internally earlier. In the revised manuscript we will change it to a table. To improve the readability of the table, we will also clean up in the metrics by putting some of them in supplementary materials. We are open for other suggestions for improving the readability.

**R2_C7:**
6) Why the calibration of LE (scenario A), does not improve the climatology of LE during March at all? (see Figure 4, please clarify)
**Reply:**
Actually, the calibration of LE (Scenario A, D and E) does slightly improve the climatology of LE during March compared to the control run (Fig 4a).

**R2_C8:**
7) How is it possible that the calibrated sand and clay values have such a large spread among scenarios? Sand[%] and Clay[%] could probably be well estimated by field measurements which you have available. I would instead calibrate some parameters which can not be measured in the field.
**Reply:**
We are interested in estimating the hydraulic properties of the soil in the form of the retention and hydraulic conductivity functions. However, in CLM5 it is not possible to specify these functions directly. Instead, %sand and %clay are used for estimating the Clapp-Hornberger exponent B and therefore we consider %sand and %clay as calibration parameters regardless of the values they may have from field measurements.

**R2_C9:**
8) It might also be interesting to see the scenarios aggregated into monthly seasonal values in addition to the diurnal climatology.
**Reply:**
We certainly take this as a legitimate suggestion. However, to constrain the study we have chosen to focus on the diurnal variations and not the seasonal values. As our observation data is exclusive in the way that we have hourly observations available for a long time period, we choose to focus on the diurnal variations.

**R2_C10:**
Data availability: under the provided link, data can not be easily found. Also, the processing codes are not available.
**Reply:** If interested parties need help in locating the data at the provided link the corresponding author can assist in this. The processing codes can also be made available by the corresponding author. This will be stated in the manuscript.

**R2_C11:**
Textual suggestion:

Page 2, Line 13: practice is to use => practice to use

Page 3, Line 13: list of LSM is too short, why not be more extensive here, include some more operationally used LSMs.
**Reply:** The list of LSMs will be expanded in the revised manuscript.

Page 3, Line 25: correct parenthesis around the reference.

Page 4 Line 6: few => a few

Page 4 Line 13: observations are available => observations available

Page 4 Line 16: combine => combines

Page 5 Line 16: were => was

Page 5 Line 23: of => between

Page 6 Line 11: reach => reaches

Page 6 Line 24: leaf => leaves

Other textual English improvements should be double checked as well.

**Reply:** All these suggestions will be corrected

---

## Author Comment (AC3)

**Reviewer (R3) specific comments (C)**

**R3_C1:**
Overall, Denager et al presents an interesting and impressive study, involving a complex model (CLM), top-level observations (especially regarding the water balance) and an advanced calibration scheme for optimizing the model parameters. Yet, the results are somewhat contradictory, and the manuscript should be improved before acceptance.

I also find it a somewhat remarkable result, that such elaborate setups and multitude of observations are needed to improve the model performance. To me, it points rather to issues in the model (unless all issues can be blamed on the lack of energy balance closure). Given the text in the introduction, and especially the highly relevant quote by Clark et al., I question whether the approach of keeping the highly complicated LSMs and needing to perform elaborate model calibrations (involving a large number of observations really) really is a good way forward for the community. It would be interesting if the authors could comment on this aspect in the study.
**Reply:** We will include a new paragraph in the discussion section; "Model physics in land surface models" to discuss these aspects. See paragraph below:

**Model physics in land surface models:** In this study we have shown that applying the model with observed parameter values were possible (Scenario Z) did not lead to an improved model performance (compared to Scenario X, which use look-up table parameter values), which can potentially be interpreted as deficiencies in the model physics. As stated by Clark et al. (2015) more and more advanced descriptions of the processes have been built into LSM codes. This induces increased model complexity and expands the associated number of parameters in the model equations (Mendoza et al., 2014). Parameter optimization in complex models is complicated, and there is a possibility that LSMs may not be parameterized appropriately. Several authors have contested the complexity of LSMs (Clark et al., 2015; Franks et al., 1999; McCabe et al., 2005; Williams et al., 2009) and suggested a reassessment of the structure and process representations. An overall simplification of the LSMs would enable a more profound parameter optimization and utilization of measured data. This would lead to more parsimonious LSMs and utilizing the well-establish model evaluation within hydrology considering uncertainties in data, model parameters and conceptual understanding (Refsgaard et al., 2021), would enhance the model evaluation of LSMs. In this way, the hydrology and LSM modelling communities could benefit even more from each other (Clark et al., 2015).

**R3_C2:**
Results and main conclusion: The many small tables of Figure 1 are hard to read and also hard to interpret.
**Reply:** See R2_C6

**R3_C3:**
I assume that it is the results in these summary tables that lead the authors to their main conclusion "that mathematical regularization is a compelling method to improve current practice of using look up tables to define parameter values in LSMs" (a similar claim is made in the first paragraph of the Discussion, page 19).

The authors should explain clearly how they reach this conclusion, since their approach also shows obvious weaknesses. Compared to the control, several error metrices increase when applying the optimization. A further example is that they can only demonstrate improvement by letting observed soil content properties drift away from their observations values (page 17, lines 9-23). Doesn't this rather point to a need for improving the model physics?
**Reply:** We do not agree that several error metrics increase when applying the optimization. Only 8 out of 80 error metrics included in the optimization in scenario A-H increase

compared to the control run (Scenario X). However, we agree that there are outcomes of the study which point to a need for improving the model physics. We will elaborate on this is the discussion of the revised manuscript.

**R3_C4:**
Another conclusion (lines 13-14, page24) is that use of soil moisture data in the optimization improved soil water storage modeling. Isn't this a rather expected result? Maybe a quantification of this improvement would be more relevant, or a comment on how the model physics could potentially be improved for sites as well as a comment on what to do for the vast majority of sites where such elaborate measurements are not present.
**Reply:** Yes this is an expected result, and it shows that uncalibrated models are not able to match observed absolute soil moisture. The quantification of the improvement of soil moisture is shown in figure 1 and figure 3.

**R3_C5:** The authors considerable emphasis on the question whether LE or H is the main culprit when it comes to the lack of energy balance closure (Conclusions, lines 18-20). They highlight that their results point to H, but their site appears to have many more observations related to the water budget than for the heat balance. Neither air nor soil temperature is used in any of the calibration scenarios. Could this result not have been the complete opposite, if they had instead focused on the heat budget and neglected to include all the soil water and moisture parameters? My recommendation is to treat this result with more caution, and at least remove it from the abstract. Rather, the authors should highlight other advantages of their results, for example the relevant conclusion stated on lines 21-25 of the Conclusion.
**Reply:** The air temperature is an input to CLM5 and is therefore already included in the model setup. We agree that the conclusion on the bias on the simulated H should be taken with caution. As suggested, we have removed it from the abstract.

In response to the concern raised by the reviewer that the heat budget is less constrained than the water budget we have expanded the study by two additional scenarios that include soil temperature as a target variable. The results of scenario E2 and H2 (including Tsoil) are only slightly different from scenario E and H (not including Tsoil). In the revised manuscript scenario E2 and H2 will get other names, so that all the scenario names are in alphabetic order.

A slight improvement in $ME_H$ from -11 $Wm^{-2}$ to -7.9 $Wm^{-2}$ is obtained from scenario E to E2, but $ME_H$ is still highly biased, and the remaining metrics for H are not improved when including Tsoil as target variable.

**Scenario E2 (H and Tsoil)**

| | ?? | ME | MAE | RMSE | NSE | r |
|---|---|---|---|---|---|---|
| LE | 81 | -2.6 | 18 | 34 | 0.65 | 0.88 |
| H | 84 | -7.9 | 22 | 33 | 0.55 | 0.77 |
| q | 107 | 0.006 | 0.050 | 0.10 | -0.00 | 0.44 |
| SWC | 489 | -0.059 | 0.064 | 0.07 | -2.19 | 0.84 |
| Ts | 72 | -7.9 | 15 | 47 | 0.01 | 0.20 |
| Sout | 93 | -0.6 | 8 | 16 | 0.77 | 0.88 |
| Rn | 98 | 0.4 | 15 | 23 | 0.97 | 0.99 |

**Scenario H2 (H, q, SWC and Tsoil)**

| | ?? | ME | MAE | RMSE | NSE | r |
|---|---|---|---|---|---|---|
| LE | 75 | 0.1 | 17 | 32 | 0.68 | 0.87 |
| H | 85 | -10.1 | 22 | 33 | 0.54 | 0.78 |
| q | 88 | 0.002 | 0.046 | 0.09 | 0.18 | 0.52 |
| SWC | 50 | -0.000 | 0.016 | 0.02 | 0.67 | 0.83 |
| Ts | 71 | -8.0 | 15 | 47 | 0.01 | 0.20 |
| Sout | 93 | -0.5 | 8 | 16 | 0.77 | 0.88 |
| Rn | 101 | 1.0 | 15 | 23 | 0.97 | 0.99 |

*Figure 1: Summary statistics for additional scenarios E2 and H2. Blue color indicate that the variable were included in the calibration for the given scenario.*

**R3_C6:** In agreement with another reviewer, I think that the manuscript would gain in value if the authors focused more on how to choose target calibration variables, and what the presented results tell could us in terms of method applicability and generality.

**Reply:**
The abstract will be rewritten (see R2_C3) and a paragraph on "Model physics in land surface models" included in the discussion (See R3_C1).

MINOR

**R3_C7:** Regarding conservation of energy (Eq 1). The equation should at least include the heating of the surface, the top soil and the air, which must be included in any land surface model.
**Reply:** As in standard eddy covariance studies, Eq. 1 neglect minor fluxes and storage terms. However, the heating of surface, the top soil and the air is included in CLM5.

**R3_C8:** Numerous places in the paper the term "physical laws" are mentioned and these should be replaced with the precise terms. Which physical laws are, for example, used to simulate H and LE (page 7, row 3)? LSMs typically apply parameterizations including many parameters.
**Reply:** CLM5 use the Monin-Obukhov similarity theory to simulate H and LE. This is stated the manuscript. We agree that the term "physical laws" should be changed to Monin-Obukhov similarity theory where relevant.

**R3_C9:** Table 1: In two places, the percentage value of sand content exceeds 100% indicating that the parameter values have not been properly bounded.
**Reply:** We are certainly aware of this problem and have already commented on this in the manuscript and in R2_C8.

**R3_C10:** Appendix A: The columns for X and Z appear to be swapped.
**Reply:** We have double-checked, and the columns are not swapped.

**Reviewer:** Appendix A: The authors claim that the site is homogeneous, which means that the measured LAI on the site is valid for the whole footprint of the EC flux observations. The scenarios based including LE tends to yield larger values for LAI compared to the observed values, which could indicate the presence of photosynthesizing plants in the footprint of the EC observations. There are very few sites that can be characterized as being completely homogeneous, and all inhomogeneities add to the mismatch between the model world and the real-world situation.

**Reply:** We agree, no sites can be claimed as homogeneous and we will thus remove this statement.